# Measurement Accuracy and Repeatability of RECIST-Defined Pulmonary Lesions and Lymph Nodes in Ultra-Low-Dose CT Based on Deep Learning Image Reconstruction

**DOI:** 10.3390/cancers14205016

**Published:** 2022-10-13

**Authors:** Keke Zhao, Beibei Jiang, Shuai Zhang, Lu Zhang, Lin Zhang, Yan Feng, Jianying Li, Yaping Zhang, Xueqian Xie

**Affiliations:** 1Radiology Department, Shanghai General Hospital, Shanghai Jiao Tong University School of Medicine, Haining Rd.100, Shanghai 200080, China; 2CT Imaging Research Center, GE Healthcare China, Shanghai 201203, China

**Keywords:** lung cancer, pulmonary nodule, artificial intelligence, Response Evaluation Criteria in Solid Tumors, dimensional measurement accuracy, computed tomography

## Abstract

**Simple Summary:**

This study compared the measured diameters of Response Evaluation Criteria in Solid Tumors (RECIST)-defined chest target lesions and lymph nodes between deep learning image reconstruction (DLIR)-based ultra-low-dose CT (ULDCT) and contrast-enhanced CT and found that the measured diameters in ULDCT were highly correlated with that of contrast-enhanced CT and highly repeatable. It is hopeful to evaluate pulmonary lesions, nodules, and lymph nodes of different sizes by using ULDCT in the future, as it is beneficial to repeated scanning in tumor response evaluation and lung cancer screening. ULDCT is expected to further reduce the radiation dose of chest imaging.

**Abstract:**

Background: Deep learning image reconstruction (DLIR) improves image quality. We aimed to compare the measured diameter of pulmonary lesions and lymph nodes between DLIR-based ultra-low-dose CT (ULDCT) and contrast-enhanced CT. Methods: The consecutive adult patients with noncontrast chest ULDCT (0.07–0.14 mSv) and contrast-enhanced CT (2.38 mSv) were prospectively enrolled. Patients with poor image quality and body mass index ≥ 30 kg/m^2^ were excluded. The diameter of pulmonary target lesions and lymph nodes defined by Response Evaluation Criteria in Solid Tumors (RECIST) was measured. The measurement variability between ULDCT and enhanced CT was evaluated by Bland-Altman analysis. Results: The 141 enrolled patients (62 ± 12 years) had 89 RECIST-defined measurable pulmonary target lesions (including 30 malignant lesions, mainly adenocarcinomas) and 45 measurable mediastinal lymph nodes (12 malignant). The measurement variation of pulmonary lesions between high-strength DLIR (DLIR-H) images of ULDCT and contrast-enhanced CT was 2.2% (95% CI: 1.7% to 2.6%) and the variation of lymph nodes was 1.4% (1.0% to 1.9%). Conclusions: The measured diameters of pulmonary lesions and lymph nodes in DLIR-H images of ULDCT are highly close to those of contrast-enhanced CT. DLIR-based ULDCT may facilitate evaluating target lesions with greatly reduced radiation exposure in tumor evaluation and lung cancer screening.

## 1. Introduction

Lung cancer is a malignant tumor with a high mortality rate [1]. Adenocarcinoma accounts for 40% of all lung cancer and is the most common type [2,3]. In 2020, the Dutch-Belgian lung cancer screening trial (NELSON) reported that after 10 years of follow-up, CT screening can reduce lung cancer-related mortality from 3.3 to 2.5 cases per 1000 person-years [4]. Therefore, regular CT screening is essential for the early detection of lung cancer. Besides early detection, the introduction of tumor genomic profiling and targeted therapy dramatically changed the treatment of lung cancer [5]. Accurate evaluation of tumor response is the basis of targeted therapy, which depends on repeated imaging examination and measurement [6].

Response Evaluation Criteria in Solid Tumors (RECIST) is widely used to evaluate the tumor response to treatment. RECIST represents the curative effect according to the change of lesion diameter in medical images. In the evaluation of lung cancer, the long diameter of pulmonary lesions and the short diameter of lymph nodes are the main indicators of tumor response [7]. Contrast-enhanced CT is usually used to evaluate lung cancer, metastasis, and lymph nodes [8], but its risk of ionizing radiation and contrast medium nephrotoxicity remains controversial [9]. Since the evaluation of tumor response requires repeated CT scans, it is necessary to develop new technologies to reduce radiation exposure and contrast medium.

Artificial intelligence (AI) promotes the development of medical imaging in many ways, such as image analysis, image quality improvement, and image reconstruction. The typical applications of AI in pulmonary imaging are cancer detection, characterization, classification [10,11], and lung cancer screening [12,13,14]. As one of the emerging technologies of AI, deep learning image reconstruction (DLIR) has accelerated the development of ultra-low-dose CT (ULDCT) by improving the signal-to-noise ratio, contrast-to-noise ratio, and lung nodule detection rate [15]. At present, an iterative reconstruction (IR) algorithm is commonly used for image reconstruction, which refers to starting from an image assumption and comparing it with the real-time measured value while constantly adjusting until the two agree [16]. However, since the IR algorithm at a high iteration level changes image texture, it performs poorly in ULDCT images with high image noise [17,18]. DLIR uses a deep neural network to emulate the texture of standard-dose filtered back-projection (FBP) images while providing strong noise reduction and maintaining high-contrast spatial resolution [19]. DLIR can improve the image quality in many areas, such as quantitative emphysema, evaluation of abdominal tumors, and examination of mediastinal tissue [20,21,22]. Our earlier study has demonstrated that at the reduced radiation dose of chest CT to 0.07 mSv~0.14 mSv, accounting for only 5%~7% of low-dose CT and 1% of standard-dose CT, DLIR can provide acceptable image quality and a high detection rate of pulmonary nodules [23], which triggered further research on DLIR-based ULDCT.

At present, low- or standard-dose contrast-enhanced CT is widely used in evaluating chest tumors, in which the diameter-dependent RECIST is often used. To reach the principle of as low as reasonably achievable (ALARA), ULDCT is expected to be increasingly implemented in chest imaging and lung cancer screening. Although DLIR-based ULDCT has demonstrated the ability to improve image quality and maintain a high nodule detection rate, to our knowledge, ULDCT as low as 0.07 mSv~0.14 mSv has not been verified as to whether it can be used to evaluate tumor response. Therefore, we aimed to evaluate the measurement accuracy and repeatability of ULDCT based on DLIR to evaluate RECIST-defined chest target lesions, pulmonary nodules, and small lymph nodes, using contrast-enhanced CT as the reference. This study would provide innovative application of ULDCT from the perspective of lesion evaluation.

## 2. Materials and Methods

### 2.1. Patients

The consecutive adult patients from April and June 2020 were prospectively enrolled at our institute. The inclusion criteria were: (1) chest ULDCT and contrast-enhanced CT on the same day; (2) with at least one pulmonary target lesion or one lymph node with a short diameter >5 mm. The exclusion criteria were: (1) poor image quality due to respiratory or motion artifacts; (2) body mass index (BMI) ≥ 30 kg/m^2^. The patients with target lesions defined by RECIST were included. Considering the demand for measurement of small lymph nodes, the patients with small lymph nodes (5 mm ≤ short diameter < 10 mm) were also included. Since this study is a cross-sectional study on lesion measurement, we did not consider the patients’ baseline or follow-up CT examinations.

According to the BMI classification by the World Health Organization (WHO), patients with a BMI < 18.5 kg/m^2^ are considered low weight, a BMI between 18.5 kg/m^2^ and 24.9 kg/m^2^ is considered normal, and a BMI ≥ 25 kg/m^2^ is regarded as overweight [7].

The diagnosis of lung cancer was determined by the histological results through hematoxylin-eosin and immunohistochemical staining [24,25] of the surgically resected lesions. The local institutional review board approved this study and written informed consent was obtained.

### 2.2. CT Acquisition

All patients underwent a chest ULDCT scan with a 256-slice CT scanner (Revolution CT, GE Healthcare), followed by a contrast-enhanced CT scan. All patients were in a supine position with both arms raised. After inspiration, they held their breath for scanning.

The scanning parameters of nonenhanced ULDCT were as follows: helical scanning, detector collimation width 0.625 × 128 mm, pitch 0.992, tube voltage 70 kV, image reconstruction matrix 512 × 512, and image display field of view 350 mm. For ULDCT scanning, the tube current was randomly set to 20 mA or 40 mA that in advance produced an absorbed dose of 0.07 mSv or 0.14 mSv, respectively.

For contrast-enhanced CT, the tube voltage was 120 kV, and the tube current was automatically set by smart mA with a noise index of 20. Smart mA is a tube current modulation that automatically sets the tube current to achieve the desired noise index by adapting the tube current to the attenuation of the target body region [26]. The other scanning parameters were the same as those of nonenhanced ULDCT. Before enhancement scanning, 40~60 mL of contrast medium (Iopamiro 300, Bracco) was injected from the anterior cubital vein at a rate of 3.0~4.0 mL/s. The amount of contrast medium was calculated based on the body weight of 0.8 mL/kg [27]. The enhanced CT acquisition was performed with a fixed time delay of 45 s.

The absorbed dose of contrast-enhanced CT scanning was 2.4 ± 0.4 mSv. Therefore, the dose at ULDCT was 94%~97% lower than that at contrast-enhanced CT.

### 2.3. Image Reconstruction and Deep Learning Image Reconstruction

The image reconstruction kernel of three sets of ULDCT images included 80%-strength adaptive statistical iterative reconstruction-V (ASIR-V-80%), DLIR of moderate strength (DLIR-M), and DLIR of high strength (DLIR-H). The contrast-enhanced CT images were reconstructed with ASIR-V-50%. All images were reconstructed with the standard kernel, and the slice thickness and interval were 1.25 mm/1.25 mm.

DLIR (TrueFidelity^TM^, GE Healthcare) is an image reconstruction algorithm on dedicated CT scanners (Revolution CT, GE Healthcare). Specifically, the DLIR engine uses a deep neural network that can handle millions of parameters. The training is performed by generating output images based on the input sinogram obtained from the ULDCT raw dataset (Appendix A). Through a deep learning approach, the imaging features of the output images are compared with the corresponding reference images (FBP-reconstructed standard-dose CT images of the same object) to match the similarity and variability between the two sets of images in terms of image noise, noise texture, low-contrast resolution, high-contrast spatial resolution, and other high-order metrics. Then, the network parameters of the deep learning model are finetuned by embedded backpropagation based on the similarity and variability. The above processes are repeated iteratively until the output images precisely match the reference images. The training process uses many high-quality data sets, and the network is required to reconstruct clinical and phantom cases it has never seen before, including extremely rare cases designed to push the network to its limits, confirming its robustness.

### 2.4. Image Quality, RECIST-Defined Target Lesions, and Image Reading

One radiologist with 5 years of experience (B.J.) in chest imaging conducted an objective image quality analysis, and three-dimensionally segmented the entire lung tissue and upper air background in vitro using a data analysis platform (MATLAB 2020a, MathWorks) to evaluate image noise, expressed as the standard deviation of CT values in Hounsfield units (HU) of the lungs and background air in the ASIR-V-80%, DLIR-M, DLIR-H, and contrast-enhanced CT images.

According to RECIST, when lesions coalesce to form a conglomerate, a plane can be maintained that would help to obtain the maximal diameter measurements of each target lesion [7]. Target lesions are classified into measurable and nonmeasurable lesions. For measurable pulmonary lesions, RECIST stipulates measuring the longest diameter, and the longest diameter needs to be ≥ 10 mm. For measurable lymph nodes, the short diameter should be ≥15 mm. Nonmeasurable lesions are smaller, that is, pulmonary lesions with a long diameter < 10 mm or lymph nodes with a short diameter between 10 and 15 mm, and truly nonmeasurable lesions, such as pericardial effusion. In this study, the target lesions included pulmonary masses (diameter > 3 cm), pulmonary nodules (<3 cm), thick wall cavities, metastases, and enlarged lymph nodes. Pulmonary nodules were divided into solid, subsolid, and ground glass nodules.

In this study, three sets of ULDCT images with different reconstruction kernels (ASIR-V-80%, DLIR-M, and DLIR-H) were reconstructed, and the measurement accuracy was evaluated against the enhanced CT images as the reference. The lesions in this study were classified into: (1) RECIST-defined measurable pulmonary target lesions (long diameter ≥ 10 mm) [7], (2) RECIST-defined measurable lymph nodes (short diameter ≥15 mm), (3) RECIST-defined nonmeasurable pulmonary target lesions (long diameter < 10 mm), (4) RECIST-defined nonmeasurable lymph nodes (10 mm ≤ short diameter < 15 mm), and (5) small lymph nodes (5 mm ≤ short diameter < 10 mm).

An independent observer with 3 years of chest imaging experience (K.Z.) evaluated and measured all lesions, and an arbiter with 21 years of experience (X.X.) determined whether the results were usable. In this study, after randomly ordering all images, the observer measured the target lesions on ASIR-V-80%, DLIR-M, DLIR-H, and contrast-enhanced CT images, then the arbiter checked the results. The observer and arbiter used a PACS terminal (TView, Winning Health) to view and measure the diameters of pulmonary target lesions and lymph nodes.

To evaluate the intra-observer repeatability of measuring lesions on ULDCT images, the same observer (K.Z.) repeated the measurement of all lesions on DLIR-M and DLIR-H images one month after the first evaluation. To evaluate the inter-observer repeatability, another observer (B.J.) independently measured all lesions in the same way. The repeatability was expressed as an intraclass correlation coefficient (ICC) between two measurements. An ICC > 0.90 was ranked as highly repeatable [28].

### 2.5. Statistics

Continuous variables with normal distribution were expressed as mean ± standard deviation (SD). The difference in basic patient information was compared by independent sample Student *t* test or Chi-square test. The measurement correlation of lesions between ULDCT and contrast-enhanced CT was indicated by Pearson’s correlation coefficient (r). The measurement variability of lesions between ULDCT and contrast-enhanced CT was analyzed by Bland-Altman analysis, in which the 95% confidence interval (CI) was expressed as mean ± 1.96 SD. Taking age, sex, BMI, CT dose (0.07 or 0.14 mSv), lesion type (pulmonary lesion or lymph node), nodule type (solid, subsolid, or ground glass nodule), and histological result as independent variables, the factors affecting the difference in lesion measurement between ULDCT and contrast-enhanced CT were analyzed by multiple linear regression. A p-value < 0.05 was considered statistically significant. A statistical package (MedCalc 20.0, MedCalc Software Ltd., Ostend, Belgium) was used for data analysis.

## 3. Results

### 3.1. Patients

A total of 141 patients (age 62 ± 12 years) were finally enrolled, including 90 males (63.8%) and 51 females (36.2%) (Table 1). There were 151 RECIST-defined measurable target lesions in the 141 patients, including 89 (58.9%) pulmonary lesions (long diameter ≥ 10 mm) and 62 (41.1%) lymph nodes (short diameter ≥ 15 mm). The 89 pulmonary lesions included 30 malignant (19 adenocarcinomas, 3 squamous cell carcinomas, 1 adenosquamous cell carcinoma, 4 metastatic tumors, and 3 large-cell neuroendocrine carcinomas) and 59 with benign or without histological results. The 62 lymph nodes included 17 hilar and 45 mediastinal lymph nodes. The 45 mediastinal lymph nodes included 12 malignant (3 adenocarcinomas, 1 squamous carcinoma, 1 adenosquamous carcinoma, 1 metastatic tumor, 1 small cell carcinoma, and 5 large cell neuroendocrine carcinomas), and 33 benign or without histological results.

There were 277 nonmeasurable target lesions in the 141 patients, including 206 pulmonary lesions (5 mm ≤ long diameter <10 mm) and 71 lymph nodes (10 mm ≤ short diameter < 15 mm). The 206 pulmonary lesions included 96 solid, 78 subsolid, and 32 ground glass nodules. The 206 lesions included 7 malignant (5 adenocarcinomas, 1 squamous carcinoma, and 1 small cell carcinoma), and 199 benign or without histological results. The 71 nonmeasurable lymph nodes included 8 hilar and 63 mediastinal lymph nodes. The 63 mediastinal lymph nodes included 9 malignant (6 adenocarcinomas, 1 squamous carcinoma, and 2 large cell neuroendocrine carcinomas) and 54 benign or without histological results (Figure 1).

There were 64 small lymph nodes (5 mm ≤ short diameter < 10 mm) in the 141 patients, including 4 malignant and 60 benign or without histological results.

### 3.2. Image Quality, Lesion Measurement, and Measurement Repeatability

The mean image noise of lung tissue was 53HU ± 4, 54HU ± 4, 51HU ± 4, and 46HU ± 4 for the three sets of ULDCT images (ASIR-V-80%, DLIR-M, and DLIR-H) and contrast-enhanced CT images, respectively. The mean image noise of air background was 29HU ± 4, 27HU ± 4, 23HU ± 4, and 22HU ± 5, respectively. Among the three sets of ULDCT images, DLIR-H had the lowest lung tissue noise and background noise.

For RECIST-defined pulmonary target lesions, the correlation coefficient of measured diameter between ULDCT and contrast-enhanced CT was 0.999 (95% CI: 0.998 to 0.999), 0.998 (0.997 to 0.999), and 0.999 (0.999 to 1.000) for ASIR-V-80%, DLIR-M, and DLIR-H images, respectively. For mediastinal lymph nodes, the correlation coefficient was 0.997 (0.995 to 0.999), 0.997 (0.995 to 0.998), and 0.999 (0.998 to 1.000), respectively. For hilar lymph nodes, the correlation coefficient was 0.993 (0.979 to 0.997), 0.995 (0.984 to 0.998), and 0.997 (0.991 to 0.998), respectively. The measured diameters of target lesions in ULDCT were highly correlated with those in contrast-enhanced CT (Table 2, Figure 1 and Figure 2).

For pulmonary target lesions, Bland-Altman analysis showed a variation of 4.6% (95% CI: 3.9% to 5.3%), 4.7% (3.9% to 5.4%), and 2.2% (1.7% to 2.6%) of measured diameter for ASIR-V-80%, DLIR-M, and DLIR-H images against that in contrast-enhanced CT, respectively. For mediastinal lymph nodes, the variation was 3.4% (2.5% to 4.2%), 4.0% (3.1% to 4.9%), and 1.4% (1.0% to 1.9%), respectively. For measurable hilar lymph nodes, the variation was 5.0% (2.5% to 7.4%), 3.9% (1.8% to 6.0%), and 2.3% (0.6% to 3.9%), respectively. Among the three sets of ULDCT images, DLIR-H showed the lowest variability (Table 3, and Figure 3).

For RECIST-defined nonmeasurable pulmonary target lesions, the correlation coefficient of measured diameter between ULDCT and contrast-enhanced CT was 0.977 (0.970 to 0.982), 0.987 (0.983 to 0.990), and 0.995 (0.994 to 0.996) for ASIR-V-80%, DLIR-M, and DLIR-H images, respectively. For mediastinal lymph nodes, the correlation coefficient was 0.937 (0.898 to 0.962), 0.939 (0.901 to 0.963), and 0.970 (0.951 to 0.982), respectively. For nonmeasurable hilar lymph nodes, the correlation coefficient was 0.994 (0.966 to 0.999), 0.969 (0.835 to 0.995), and 0.997 (0.982 to 0.999), respectively. The measured diameters of nonmeasurable target lesions in ULDCT were also highly correlated with those in contrast-enhanced CT (Table 4, Figure 4 and Figure 5).

For nonmeasurable pulmonary target lesions, Bland-Altman analysis showed a variation of 5.7% (4.7% to 6.7%), 5.1% (4.3% to 5.8%), and 2.2% (1.7% to 2.6%) for ASIR-V-80%, DLIR-M, and DLIR-H images against that in contrast-enhanced CT, respectively. For mediastinal lymph nodes, the variation was 6.1% (5.2% to 7.1%), 6.4% (5.4% to 7.4%), and 2.9% (2.2% to 3.5%), respectively. For hilar lymph nodes, the variation was 5.2% (4.3% to 6.1%), 6.4% (4.4% to 8.4%), and 3.2% (2.4% to 4.0%), respectively. Among the three reconstruction image sets, DLIR-H still showed the lowest variability (Table 5).

The intra-observer ICC of the DLIR-M images was 0.996 (95% CI: 0.993 to 0.997), 0.998 (0.997 to 0.999), 0.978 (0.953 to 0.988), 0.933 (0.895 to 0.958), and 0.961 (0.900 to 0.981) for measurable pulmonary lesions, measurable lymph nodes, nonmeasurable pulmonary lesions, nonmeasurable lymph nodes, and small lymph nodes, respectively. The intra-observer ICC of DLIR-H was 0.996 (0.993 to 0.998), 0.998 (0.995 to 0.999), 0.986 (0.935 to 0.994), 0.960 (0.843 to 0.984), and 0.942 (0.804 to 0.975), respectively. The inter-observer ICC of DLIR-M was 0.995 (95% CI: 0.983 to 0.998), 0.992 (0.975 to 0.996), 0.954 (0.938 to 0.966), 0.840 (0.437 to 0.935), and 0.918 (0.846 to 0.954), respectively. The inter-observer ICC of DLIR-H was 0.997 (0.965 to 0.999), 0.993 (0.970 to 0.997), 0.931 (0.632 to 0.974), 0.910 (0.345 to 0.971), and 0.835 (0.215 to 0.943), respectively. Therefore, the measurement of target lesions in ULDCT was highly repeatable.

### 3.3. Association with Influential Factors

For measurable target lesions, multiple linear regression showed no significant association (all *p* > 0.05) between the measurement difference (between DLIR-H and contrast-enhanced CT images) and potential factors (age, sex, BMI, CT dose, lesion type, and histological result), and the same for ASIR-V-80% and DLIR-M images (all *p* > 0.05) (Table 6).

For nonmeasurable pulmonary lesions and lymph nodes, multiple linear regression also showed no significant association (all p>0.05) between the measurement difference and potential factors (age, sex, BMI, CT dose, nodule type, and histological result) for ASIR-V-80%, DLIR-M, and DLIR-H (all p>0.05) (Table 7 and Table 8).

## 4. Discussion

This study evaluated the measurement accuracy and repeatability of pulmonary lesions and lymph nodes in DLIR-based ULDCT, including benign and malignant lesions mainly represented by lung adenocarcinoma. The measured diameters of lesions in ULDCT images were highly correlated with that in contrast-enhanced CT (r = 0.991~0.999). For measurable pulmonary lesions and lymph nodes, Bland-Altman analysis showed that the measurement variability between DLIR-H and contrast-enhanced CT was only 1%~2%. These findings provoke the use of ULDCT in evaluating chest lesions at a greatly reduced radiation dose.

The major advantage of ULDCT is that it can greatly reduce the radiation exposure risk of patients when evaluating chest lesions. The National Comprehensive Cancer Network (NCCN) guideline has defined that the radiation exposure of standard-dose chest CT is 7 mSv, and that of low-dose CT is 1.5 mSv [29]. Rampanelli et al. estimated through mathematical models that after 10 years of screening, the cumulative effective dose of each subject was 9.3 mSv~13 mSv, which may increase the risk of radiation-induced malignancy [30]. ALARA is the guiding principle for controlling radiation exposure in imaging examination. Based on the hardware configuration of low tube voltage and current and software solution of DLIR, our study obtained high-quality ULDCT (0.07 mSv~0.14 mSv) images, which were 99% lower than standard-dose CT and 91%~95% lower than low-dose CT. This study also discussed the feasibility of evaluating tumor lesions without using contrast medium, thus avoiding the use of iodine-containing contrast medium with potential nephrotoxicity. By using ULDCT in this study, the risk of radiation and nephrotoxicity during tumor evaluation can be reduced.

Recent technological advances of ULDCT have led doctors to improve their practice of diagnosis and evaluation [31,32,33]. Similarly to this study, other studies have reported improvements in image quality of CT scans of other body parts by DLIR. Parakh et al. studied DLIR images in visualizing abdominal tumors and found that, compared with the ASIR-V, DLIR had lower image noise and a higher contrast-to-noise ratio; thus, it can improve the detection rate of abdominal tumors [34]. Noda et al. evaluated the image quality and detectability in low-dose portal-venous-phase CT based on DLIR and found that it can reduce the radiation exposure by 80% compared with that in standard-dose CT, while maintaining image quality and lesion detection rate [35]. Kim et al. found that the signal-to-noise ratio of DLIR-H was 30% higher than that of ASIR-V, and DLIR-H showed higher lesion detection ability regardless of lesion location [36]. In our study, the measured lesion diameters in ULDCT were highly correlated with those in contrast-enhanced CT. Bland-Altman analysis showed that among the three reconstruction methods (ASIR-V-80%, DLIR-M, and DLIR-H), DLIR-H had the smallest difference in measured lesion diameters between ULDCT and contrast-enhanced CT. For pulmonary lesions and mediastinal lymph nodes, the difference in measured diameter was only 2.2% and 1.4%, respectively. When evaluating lymph nodes, the measurement variability in hilar lymph nodes was slightly larger than that of mediastinal lymph nodes, which may have been due to the influence of surrounding hilar tissues [37].

Compared with ASIR-V, the measured lesion diameters in DLIR-H images were closer to that of contrast-enhanced CT. The reason may be that DLIR-H performs better in decreasing image noise, reducing streak artifacts, and improving overall image quality [38]. However, the measured diameters between DLIR and enhanced-CT images are not completely consistent. The variance could be due to the higher edge-rise distance and attenuation skewness of reconstructed images, which result in decreased spatial resolution and increased image distortion [22]. Nam et al. found that although distortion artifacts caused by DLIR were more common in reconstructed images, only 0.3% of cases were considered difficult to diagnose [39]. To sum up, DLIR is a promising image reconstruction algorithm, which can maintain the accuracy of lesion measurement at a reduced radiation dose. It is hopeful that ULDCT combined with DLIR-H will be widely used in clinical practice.

This study has limitations. First, this study is a single-center study, and the sample size is not large. Although this study provides promising results, there may be some deviations from large-scale multicenter studies. A future multicenter study is encouraged to strengthen the conclusion and expand the application of ULDCT. Second, we used the DLIR technology from one vendor. Currently, multiple DLIR algorithms are available. It is necessary to validate the findings of this study with multiple algorithms. Third, due to the high image noise of obese patients, this study did not include obese patients. Further technical research is needed to develop ULDCT scanning and image processing technology for obese patients. Fourth, since the number of lesions with histological results was limited, it is meaningful to continue to explore the accuracy and repeatability of measured diameters in lesions with different histological results.

## 5. Conclusions

This study compared the measured diameters of pulmonary lesions and lymph nodes in DLIR-based ULDCT and contrast-enhanced CT. It was found that ULDCT at 0.07~0.14 mSv was highly close to that of contrast-enhanced CT, and the measurement repeatability was high. Because DLIR-based ULDCT greatly reduces the radiation dose and avoids contrast medium, it provides a promising method to evaluate pulmonary tumors, nodules, and lymph nodes of different sizes, which is conducive to repeated scanning in lung cancer screening and tumor response evaluation, such as in lung adenocarcinoma. ULDCT can help to reduce the risks of radiation exposure and may reach as low as reasonably achievable (ALARA) in the evaluation of chest lesions in the future.

## Figures and Tables

**Figure 1 cancers-14-05016-f001:**
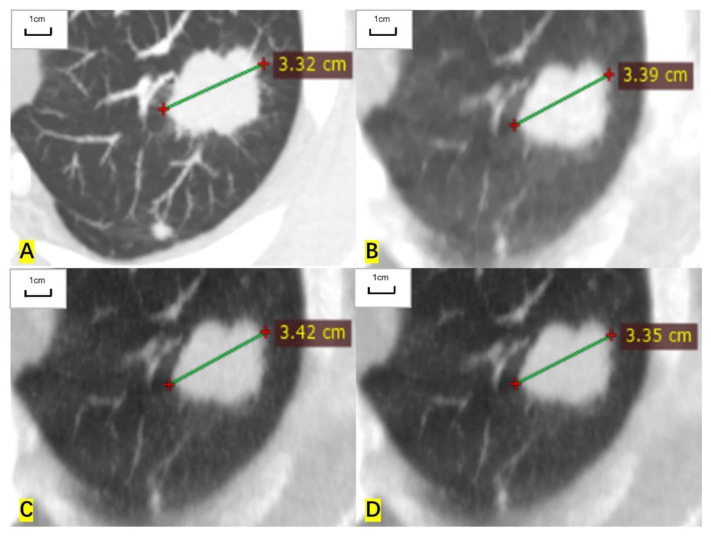
A 57-year-old man had a measurable target lesion in the upper lobe of the left lung and the histological result was lung adenocarcinoma. (**A**) The long diameter measured on the contrast-enhanced CT image is 33.2 mm. (**B**) The measured long diameter on the ASIR-V-80% reconstructed image is 33.9 mm. (**C**) The measured long diameter on the DLIR-M image is 34.2 mm. (**D**) The measured long diameter on the DLIR-H images is 33.5 mm. The long diameter is overestimated by 2.1%, 3.0%, and 0.9%, respectively, compared with that of contrast-enhanced CT.

**Figure 2 cancers-14-05016-f002:**
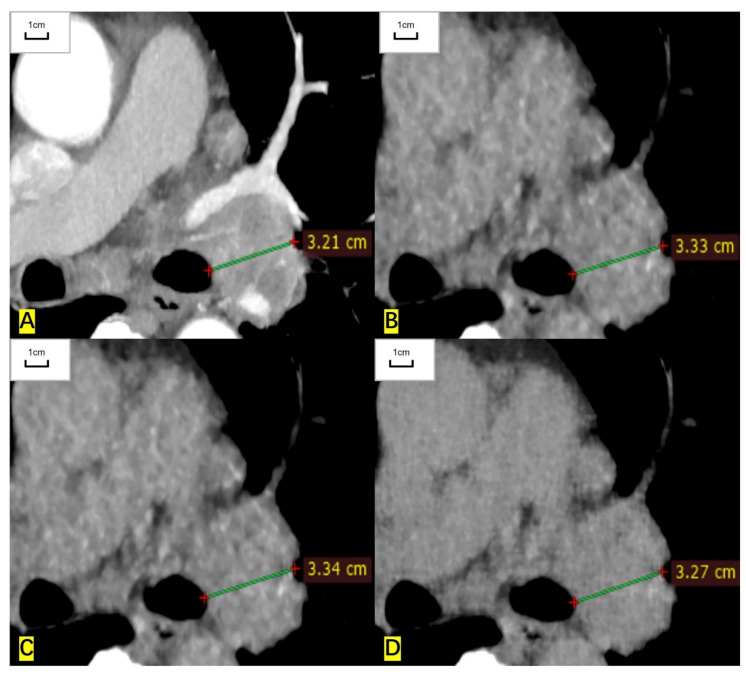
A 75-year-old woman had a lymph node in the left hilum and the histological result was adenocarcinoma. (**A**) The short diameter measured on the contrast-enhanced CT image is 32.1 mm. (**B**) The measured short diameter on the ASIR-V-80% reconstructed image is 33.3 mm. (**C**) The measured short diameter on the DLIR-M image is 33.4 mm. (**D**) The measured short diameter on the DLIR-H images is 32.7 mm. The short diameter is overestimated by 3.7%, 4.0%, and 1.9%, respectively, compared with that of contrast-enhanced CT.

**Figure 3 cancers-14-05016-f003:**
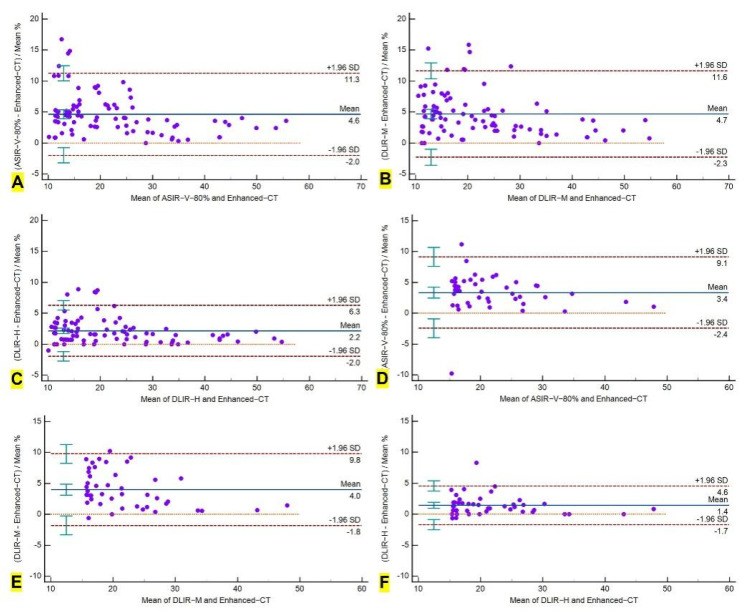
Bland-Altman analysis of the variability of measured diameters of measurable target lesions on ultra-low-dose CT and contrast-enhanced CT images. (**A**–**C**) Mean diameter difference of pulmonary target lesions between ASIR-V-80% and contrast-enhanced CT is 4.6% (95% CI: 3.9% to 5.3%), between DLIR-M and contrast-enhanced CT is 4.7% (3.9% to 5.4%), and between DLIR-H and contrast-enhanced CT is 2.2% (1.7% to 2.6%). (**D**–**F**) Mean diameter difference of measurable mediastinal lymph nodes between ASIR-V-80% and contrast-enhanced CT is 3.4% (2.5 to 4.2%), between DLIR-M and contrast-enhanced CT is 4.0% (3.1 to 4.9%), and between DLIR-H and contrast-enhanced CT is 1.4% (1.0 to 1.9%).

**Figure 4 cancers-14-05016-f004:**
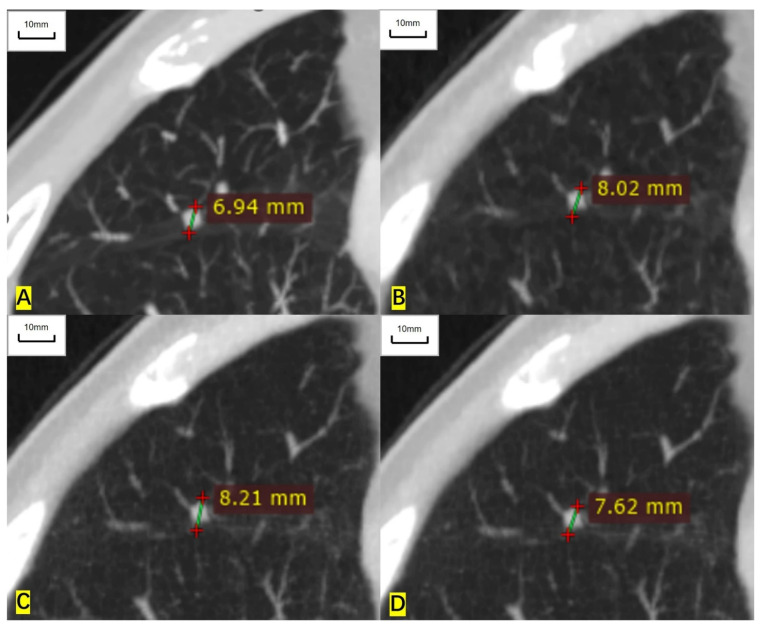
A 64-year-old man had a solid nodule in the middle lobe of the right lung and the histological result was lung adenocarcinoma. (**A**) The long diameter measured on the contrast-enhanced CT is 6.94 mm. (**B**) The long diameter measured on the ASIR-V-80% image is 8.02 mm, overestimated by 15.6% compared with that of contrast-enhanced CT. (**C**) The long diameter measured on the DLIR-M image is 8.21mm, overestimated by 18.3%. (**D**) The long diameter measured on the DLIR-H images is 7.62 mm, overestimated by 9.8%.

**Figure 5 cancers-14-05016-f005:**
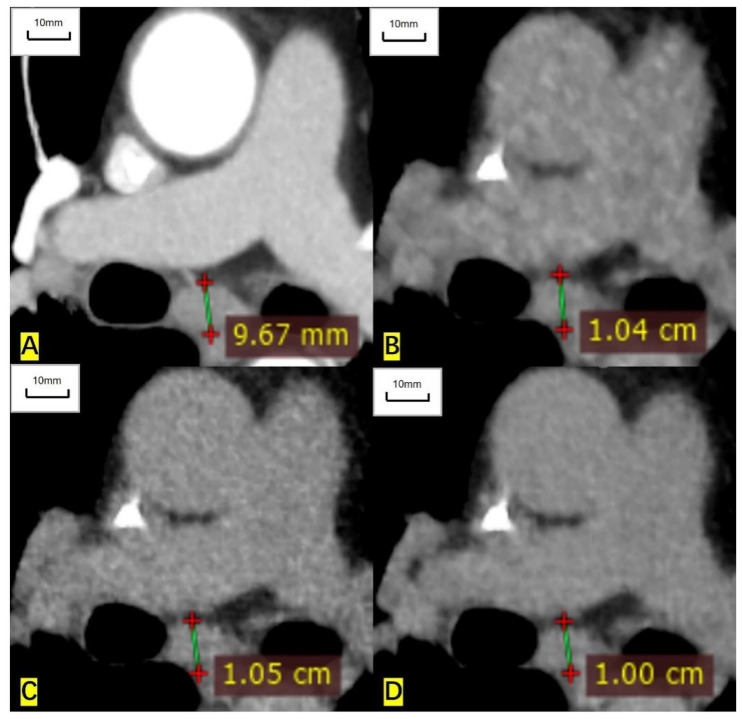
A 65-year-old man had an enlarged subcarinal lymph node, and the histological result was unavailable. (**A**) The short diameter measured on the contrast-enhanced CT image is 9.67 mm. (**B**) The measured short diameter on the ASIR-V-80% reconstructed image is 10.4 mm. (**C**) The measured short diameter on the DLIR-M image is 10.5 mm. (**D**) The measured short diameter on the DLIR-H image is 10.00 mm. The short diameter is overestimated by 7.5%, 8.6%, and 3.4%, respectively, compared with that of contrast-enhanced CT.

**Table 1 cancers-14-05016-t001:** Basic patient characteristics.

Variable		Ultra-Low Dose CT		*p* Value
	Contrast-Enhanced CT (n = 141)	At 0.07 mSv (n = 75)	At 0.14 mSv (n = 66)	
Age/year	62 ± 12	62 ± 11	62 ± 13	0.704
Gender/n (%)				
Male Female	90 (63.9%) 51 (36.2%)	47 (33.3%) 28 (19.9%)	43 (30.5%) 23 (16.3%)	0.759
BMI/(kg·m^2^)	22.79 ± 2.94	23.02 ± 3.07	22.51 ± 2.78	0.362
<18.5	9	5	4	0.367
≥18.5 and <25	106	53	53	
≥25	26	17	9	
Measurable pulmonary lesions				
Malignant	30	15	15	0.821
Benign or no histological result	59	31	28	
Measurable lymph nodes				
Malignant	21	14	7	0.534
Benign or no histological result	41	24	17	
Nonmeasurable pulmonary lesions				
Malignant	7	4	3	0.994
Benign or no histological result	199	114	85	
Nonmeasurable lymph nodes				
Malignant	10	3	7	0.025
Benign or no histological result	61	41	20	

Note. A p-value < 0.05 is considered statistically significant.

**Table 2 cancers-14-05016-t002:** Pearson’s correlation coefficients of measured diameters of RECIST-defined measurable target lesions between ultra-low-dose CT and contrast-enhanced CT.

	Correlation Coefficient (95% CI)		
	ASIR-V-80% and Enhanced CT	DLIR-M and Enhanced CT	DLIR-H and Enhanced CT
All pulmonary target lesions	0.999 (0.998 to 0.999)	0.998 (0.997 to 0.999)	0.999 (0.999 to 1.000)
Malignant	0.998 (0.997 to 0.999)	0.998 (0.995 to 0.999)	0.999 (0.999 to 1.000)
Benign or no histological result	0.999 (0.998 to 0.999)	0.998 (0.997 to 0.999)	0.999 (0.999 to 1.000)
Mediastinal lymph nodes	0.997 (0.995 to 0.999)	0.997 (0.995 to 0.998)	0.999 (0.998 to 1.000)
Malignant	0.998 (0.991 to 0.999)	0.991 (0.968 to 0.998)	0.997 (0.989 to 0.999)
Benign or no histological result	0.997 (0.995 to 0.999)	0.998 (0.996 to 0.999)	1.000 (0.999 to 1.000)
Hilar lymph nodes	0.993 (0.979 to 0.997)	0.995 (0.984 to 0.998)	0.997 (0.991 to 0.998)

Note: RECIST = Response Evaluation Criteria in Solid Tumors; ASIR-V-80% = adaptive statistical iterative reconstruction with an 80% strength level; DLIR = deep learning image reconstruction; DLIR-M = DLIR of moderate strength; DLIR-H = DLIR of high strength.

**Table 3 cancers-14-05016-t003:** Bland-Altman analysis of measured diameter of RECIST-defined measurable target lesions between ultra-low-dose CT and contrast-enhanced CT.

	Arithmetic Mean (95% CI)		
	ASIR-V-80% and Enhanced CT	DLIR-M and Enhanced CT	DLIR-H and Enhanced CT
All pulmonary target lesions	4.6% (3.9–5.3%)	4.7% (3.9–5.4%)	2.2% (1.7–2.6%)
Malignant	3.5% (2.6–4.4%)	4.0% (2.8–5.1%)	1.7% (1.1–2.4%)
Benign or no histological result	5.2% (4.3–6.2%)	5.0% (4.0–6.0%)	2.4% (1.8–2.9%)
Mediastinal lymph nodes	3.4% (2.5–4.2%)	4.0% (3.1–4.9%)	1.4% (1.0–1.9%)
Malignant	3.5% (2.3–4.7%)	4.1% (2.0–6.2%)	2.1% (0.7–3.5%)
Benign or no histological result	3.3% (2.1–4.5%)	4.0% (3.0–5.0%)	1.2% (0.8–1.7%)
Hilar lymph nodes	5.0% (2.5–7.4%)	3.9% (1.8–6.0%)	2.3% (0.6–3.9%)

Note: 95% confidence interval (CI) is expressed as mean ± 1.96 SD. RECIST = Response Evaluation Criteria in Solid Tumors; ASIR-V-80% = adaptive statistical iterative reconstruction with an 80% strength level; DLIR = deep learning image reconstruction; DLIR-M = DLIR of moderate strength; DLIR-H = DLIR of high strength.

**Table 4 cancers-14-05016-t004:** Pearson’s correlation coefficients of measured diameters of RECIST-defined nonmeasurable lesions between ultra-low-dose CT and contrast-enhanced CT.

	Correlation Coefficient (95% CI)		
	ASIR-V-80% and Enhanced CT	DLIR-M and Enhanced CT	DLIR-H and Enhanced CT
All pulmonary target lesions	0.977 (0.970 to 0.982)	0.987 (0.983 to 0.990)	0.995 (0.994 to 0.996)
Malignant	0.997 (0.981 to 1.000)	0.999 (0.991 to 1.000)	0.997 (0.980 to 1.000)
Benign or no histological result	0.976 (0.968 to 0.982)	0.987 (0.982 to 0.990)	0.995 (0.993 to 0.996)
Solid nodules	0.961 (0.942 to 0.974)	0.987 (0.980 to 0.991)	0.996 (0.993 to 0.997)
Subsolid nodules	0.990 (0.984 to 0.993)	0.987 (0.980 to 0.992)	0.996 (0.993 to 0.997)
Ground glass nodules	0.987 (0.974 to 0.994)	0.986 (0.971 to 0.993)	0.993 (0.986 to 0.997)
Mediastinal lymph nodes	0.937 (0.898 to 0.962)	0.939 (0.901 to 0.963)	0.970 (0.951 to 0.982)
Malignant	0.960 (0.816 to 0.992)	0.965 (0.836 to 0.993)	0.974 (0.879 to 0.995)
Benign or no histological result	0.934 (0.888 to 0.961)	0.939 (0.897 to 0.965)	0.968 (0.946 to 0.982)
Hilar lymph nodes	0.994 (0.966 to 0.999)	0.969 (0.835 to 0.995)	0.997 (0.982 to 0.999)
Small lymph nodes (5 mm~10 mm)	0.945 (0.910 to 0.966)	0.961 (0.936 to 0.976)	0.976 (0.960 to 0.985)

Note: RECIST = Response Evaluation Criteria in Solid Tumors; ASIR-V-80% = adaptive statistical iterative reconstruction with an 80% strength level; DLIR = deep learning image reconstruction; DLIR-M = DLIR of moderate strength; DLIR-H = DLIR of high strength.

**Table 5 cancers-14-05016-t005:** Bland-Altman analysis of measured diameters of RECIST-defined nonmeasurable lesions between ultra-low-dose CT and contrast-enhanced CT.

	Arithmetic Mean (95% CI)		
	ASIR-V-80% and Enhanced CT	DLIR-M and Enhanced CT	DLIR-H and Enhanced CT
All pulmonary target lesions	5.7% (4.7–6.7%)	5.1% (4.3–5.8%)	2.2% (1.7–2.6%)
Malignant	4.2% (2.3–6.2%)	3.0% (1.8–4.2%)	1.1% (−0.2–2.4%)
Benign or no histological result	5.7% (4.7–6.8%)	5.1% (4.4–5.9%)	2.3% (1.7–2.7%)
Solid nodules	5.7% (3.9–7.5%)	5.0% (3.9–6.1%)	2.3% (1.6–3.0%)
Subsolid nodules	5.7% (4.6–6.8%)	5.2% (4.0–6.4%)	2.1% (1.5–2.7%)
Ground glass nodules	5.8% (3.4–8.2%)	4.8% (2.6–7.1%)	2.1% (0.4–3.9%)
Mediastinal lymph nodes	6.1% (5.2–7.1%)	6.4% (5.4–7.4%)	2.9% (2.2–3.5%)
Malignant	5.0% (3.2–6.8%)	4.3% (2.6–6.1%)	2.6% (1.1–4.0%)
Benign or no histological result	6.3% (5.2–7.4%)	6.8% (5.7–7.8%)	2.9% (2.1–3.7%)
Hilar lymph nodes	5.2% (4.3–6.1%)	6.4% (4.4–8.4%)	3.2% (2.4–4.0%)
Small lymph node (5 mm ≤ d < 10 mm)	5.7% (4.7–6.8%)	5.8% (4.9–6.7%)	2.2% (1.5–2.9%)

Note: 95% confidence interval (CI) is expressed as mean ± 1.96 SD. RECIST = Response Evaluation Criteria in Solid Tumors; ASIR-V-80% = adaptive statistical iterative reconstruction with an 80% strength level; DLIR = deep learning image reconstruction; DLIR-M = DLIR of moderate strength; DLIR-H = DLIR of high strength.

**Table 6 cancers-14-05016-t006:** Multiple linear regression analysis of the influential factors on measurement difference of measurable target lesions between ultra-low-dose CT and contrast-enhanced CT.

	ASIR−V−80% and Enhanced CT		DLIR−M and Enhanced CT		DLIR−H and Enhanced CT	
Factors	B	*p*-value	B	*p*-value	B	*p*-value
Age	0.007	0.198	0.003	0.566	0.003	0.381
Sex	0.079	0.473	−0.039	0.741	0.081	0.264
Body mass index	0.023	0.219	−0.021	0.299	−0.002	0.842
CT dose	−0.144	0.182	−0.159	0.175	−0.066	0.355
Lesion type	−0.104	0.344	−0.057	0.632	−0.044	0.540
Histological result	0.004	0.970	0.177	0.146	0.115	0.119

Note. B = regression coefficient; *p* < 0.05 is considered statistically significant. RECIST = Response Evaluation Criteria in Solid Tumors; ASIR-V-80% = adaptive statistical iterative reconstruction with an 80% strength level; DLIR = deep learning image reconstruction; DLIR-M = DLIR of moderate strength; DLIR-H = DLIR of high strength.

**Table 7 cancers-14-05016-t007:** Multiple linear regression analysis of influential factors on measurement difference of nonmeasurable pulmonary lesions between ultra-low-dose CT and contrast-enhanced CT.

	ASIR−V−80% and Enhanced CT		DLIR−M and Enhanced CT		DLIR−H and Enhanced CT	
Factors	B	*p*-value	B	*p*-value	B	*p*-value
Age	0.004	0.102	0.002	0.252	0.002	0.141
Sex	−0.028	0.590	−0.037	0.349	−0.019	0.443
Body mass index	−0.005	0.528	0.003	0.669	0.002	0.553
CT dose	−0.064	0.225	−0.046	0.246	−0.011	0.668
Nodule type	0.002	0.958	0.018	0.912	0.004	0.930
Histological result	−0.023	0.877	−0.063	0.559	−0.051	0.450

Note. B = regression coefficient; *p* < 0.05 is considered statistically significant. RECIST = Response Evaluation Criteria in Solid Tumors; ASIR-V-80% = adaptive statistical iterative reconstruction with an 80% strength level; DLIR = deep learning image reconstruction; DLIR-M = DLIR of moderate strength; DLIR-H = DLIR of high strength.

**Table 8 cancers-14-05016-t008:** Multiple linear regression analysis of influential factors on measurement difference of nonmeasurable lymph nodes between ultra-low-dose CT and contrast-enhanced CT.

	ASIR−V−80% and Enhanced CT		DLIR−M and Enhanced CT		DLIR−H and Enhanced CT	
Factors	B	*p*-value	B	*p*-value	B	*p*-value
Age	0.004	0.522	0.009	0.110	0.002	0.574
Sex	−0.006	0.964	−0.020	0.889	0.022	0.817
Body mass index	−0.005	0.810	−0.014	0.514	−0.006	0.681
CT dose	−0.071	0.598	−0.129	0.348	−0.134	0.145
Histological result	−0.089	0.632	−0.218	0.249	0.042	0.738

Note. B = regression coefficient; *p* < 0.05 is considered statistically significant. RECIST = Response Evaluation Criteria in Solid Tumors; ASIR-V-80% = adaptive statistical iterative reconstruction with an 80% strength level; DLIR = deep learning image reconstruction; DLIR-M = DLIR of moderate strength; DLIR-H = DLIR of high strength.

## Data Availability

The data presented in this study are available on request from the corresponding author.

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
