# Peer review of "Measurement Accuracy and Repeatability of RECIST-Defined Pulmonary Lesions and Lymph Nodes in Ultra-Low-Dose CT Based on Deep Learning Image Reconstruction"

_cancers, 2022, doi:10.3390/cancers14205016_

Round 1
Reviewer 1 Report (New Reviewer)
The authors have addressed the comments and suggestions properly, it reads fluent and the reasoning is building towards the conclusion.
This manuscript is a resubmission of an earlier submission. The following is a list of the peer review reports and author responses from that submission.
Round 1
Reviewer 1 Report
General Comments:
This paper aimed to compare chest target lesions on non-contrast chest ULDCT and contrast-enhanced low-dose CT in 147 patients. Four sets of reconstructed CT images (3 ULDCT images: ASIR-V-80%, DLIR-M and DLIR-H; and 1 set of contrast-enhanced CT) were performed. The authors found that the diameters of target lesions in ULDCT images highly correlated with contrast-enhanced CT. For tumor lesions, measurement variability was 2.2% between DLIR-H and contrast-enhanced CT, 4.7% for ASIR-V-80% and 4.7% for DLIR-M. For lymph nodes measurement variability was 1.4% and 2.3% less than those of ASIR-V-80% and DLIR-M, respectively. The authors argued that non-contrast ULDCT images reconstructed with DLIR-H allow repeated CT scans in patients with lung tumor at significantly lower radiation dose.
Despite the subject has major clinical interest, the study is not correctly designed, since authors compared unenhanced with enhanced CT scans; additionally, contrast medium administration is crucial for CT scans evaluation with RECIST criteria. The study has some limitations, such as the CT image quality and analysis performed by a single reader, as well as the single-centre and single-vendor nature of the study. Furthermore, the manuscript needs substantial implementation in each section.
Please see specific comments.
Specific comments:
Title: The title is informative, yet it can be improved.
Abstract:
· It does not respect Authors guidelines: it exceeds the limit of 200 words anddoes not follow the subheads Background, Methods, Results and Conclusion. Please reformat accordingly.
· The aim of the study should be better emphasized.
· Please rewrite in a more clear and schematic way the Methods section: in particular, clarify exclusion/inclusion criteria, the study protocol, and add a summary of statistical analysis performed.
· Please rewrite more clearly the Results section and please specify the lung cancer histological types of patients enrolled.
· Please rewrite the Conclusions focusing on the main findings of the study.
Key words: Some more focused key words underlining the aim of the study could be added
Introduction:
· This section is too short and needs substantial implementation, each paragraph seems to be disconnected from the others. Please explain the rationale for conducting the study based on a review of the current literature and highlight what the current study would add to the scientific field.
· Please explain both IR and DLIR methods for reconstruction of CT images.
· Please highlight the aim of the study.
Materials and Methods:
· Patients: it is not clear how the study population was enrolled, and how the lung cancer diagnosis was performed. Please clarify. Please also explain if the patients underwent baseline or follow-up CT. What did the Authors mean by “unsuccessful or incomplete image reconstruction”? Please clarify.
· CT acquisition: Please rewrite more schematically both ULDCT and enhanced CT protocol. Please explain if the enhanced CT acquisition was performed with fixed time-delay technique or the automatic bolus tracking, and provide details on how the amount of contrast medium has been calculated based.
· Image reconstruction and Deep learning image reconstruction: please merge these two subsections and rewrite them more clearly.
Image quality, RECIST-defined target lesions and Imaging reading: These subsections should be merged and rewritten more clearly as well. Manual measurement, despite the mentioned measurement of repeatability, seem too arbitrary, since differences of the magnitude of few millimeters might have significantly influenced the results. The study should be performed with automated measurement tool in order to strengthen repeatability.
Results:
· Patients: Please rewrite this subsection in a more clear and schematic way, it is difficult to understand the final population enrolled and the measurable and non-measurable lesions. Table 1 and Figure 2 should be reformatted more clearly as well.
· The following subsections should be merged similarly to what explained for Methods section and rewrite more schematically; these subsections are very difficult to read.
Discussion:
· Please rewrite the entire section and connect each paragraph more fluently.
· Please avoid personal pronouns as “we”.
· Please expand the paragraph regarding study limitations.
Conclusion: Please rewrite the section avoiding repeating the summary of the study and focusing on main results and on the possible improvements to clinical practise.
References: adequate
Tables: In general, for significant p values on tables: please highlight them in bold or with asterisks.
Figures: Please improve image quality for each figure.
Linguistic and typewriting: Linguistic improvement is necessary.
Reviewer 2 Report
The authors discuss how the deep learning image reconstruction enables ultra-low-dose CT to evaluate RECIST-defined chest target lesions. I found the methodological part to be well justified and reasonable for this type of analysis. Although the manuscript is overall well-written and structured, it might benefit from additional spell/language checking.
Title is bit confusing, kindly update it.
The introduction is deprived of the related work with the recent literature.
There are several interesting papers that look into Deep learning in healthcare. For instance, the below papers has some interesting implications that you could discuss in your Introduction and how it relates to your work.
Vulli, A.; et al.. Fine-Tuned DenseNet-169 for Breast Cancer Metastasis Prediction Using FastAI and 1-Cycle Policy. Sensors 2022, 22, 2988.
Ali, Farman, et al. "A fuzzy ontology and SVM–based Web content classification system." IEEE Access 5 (2017): 25781-25797.
Authors should further clarify and elaborate novelty in their contribution. Best is to put them in 2nd last paragraph of the introduction.
What are the key issues present study has addressed?
What are the problems faced in past and how present will address those issues?
Authors talks about deep learning, but specifically mention which algorithm they have used.
Do mention what is the pseudo code. Why authors choose particular approach and why not others? Provide references to support your claims.
What is the difference between the standard-dose CT and Ultra-low-doze CT?
Response Evaluation Criteria in Solid Tumors needs further explanation with references.
I would suggest to use some publically available dataset to validate the performance and accuracy of proposed approach.
Your proposed model needs more description, how it works in figure-1?
What are the practical implications of your research?
Conclusion is too short. Add more explanation.
What are the limitations of the present work?
Round 2
Reviewer 1 Report
General Comments:
This paper aimed to compare the evaluation of chest target lesions on non-contrast chest ULDCT and contrast-enhanced low-dose CT, in 141 patients. Four sets of reconstructed CT images (3 ULDCT images: ASIR-V-80%, DLIR-M and DLIR-H; and 1 set of contrast-enhanced CT) were performed. The authors found that the diameters of target lesions in ULDCT images highly correlated with the contrast-enhanced CT ones. The Bland-Altman analysis showed a measurement variability of 2.2% between DLIR-H and contrast-enhanced CT, less than compared to 4.7% of ASIR-V-80% and 4.7% of DLIR-M. For lymph nodes the measurement variability was 1.4% and 2.3% less than those of ASIR-V-80% and DLIR-M. The authors argued that non-contrast ULDCT images reconstructed with DLIR-H allows to repeated CT scans in patients affected by lung tumor at greatly reduced radiation dose.
Despite the subject has major clinical interest, the study is not correctly designed in my opinion, since the Authors compare unenhanced with enhanced CT scans, plus contrast medium administration is crucial for CT scans evaluation with RECIST criteria. Moreover, the patient’s enrolment is still not clear. The manuscript shows some relevant limitations and needs a real implementation.
Please see specific comments.
Specific comments:
Title: clear and informative.
Abstract:
· The Abstract does not respect the guidelines for Authors: it exceeds the limit of 200 words. Please reformat accordingly.
· Please clarify in the background paragraph the comparison with contrast enhanced CT.
· Exclusion criteria were missing in the Methods section, please add.
· Please rewrite more clearly the Results section and please specify the lung cancer histological types of patients enrolled.
Key words: adequate
Introduction:
· AI is now widely applied in field of radiological imaging and your paper was focused on the application of AI on images reconstruction (DLIR): please rewrite the paragraph about AI focusing mainly on this field. Moreover, please improve the review of the current literature (your prior study only is not sufficient) and highlight what the current study would add to the scientific field.
Materials and Methods:
· Patients:
o it is not clear how the study population was enrolled; have you enrolled all patients underwent CT, only patients with lung cancer or other? Have you considered RECIST criteria in all patients enrolled or only in patients for whom criteria were applicable. Please specify.
o it is not clear how the lung cancer diagnosis was performed (surgery? Biopsy?).
o Please also explain if the patients underwent baseline or follow-up CT.
· CT acquisition:
o Is not clear how the tube current and the amount of contrast medium has been calculated based on body size, in ULDCT and contrast-enhanced CT respectively. Have you considered only the BMI/total body weight or also lean body weight? Please clarify adding also references.
· Image reconstruction and Deep learning image reconstruction:
o please specify which AI system was applied (TrueFidelity?)
o please specify which kernel of AI reconstructed images match with standard kernel, and explain why you haven’t use the pulmonary dedicated kernels. Please rewrite more clearly.
· Image quality, RECIST-defined target lesions and Imaging reading: Please merge also the section “study group”.
Results:
· Patients: Table 1 should report also lesion measured or detected in contrat-enhanced CT, please modify.
· The following subsections should be merged similarly to what explained for Methods section and rewrite more schematically; these subsections are very difficult to read.
Discussion:
· Please rewrite more concisely the entire section; each paragraph seems to be disconnected from the others. Study results should be compared with prior studies, and similarities or differences should be extensive explore. Please consider to read the BMJ editorial (Docherty M, Smith R. The case for structuring the discussion of scientific papers. BMJ. 1999 May 8;318 (7193):12245. https://doi.org/10.1136/bmj.318.7193.1224) that suggested the correct structure for Discussion of scientific papers.
· Please rewrite more fluency the paragraph regarding study limitations.
Conclusion: adequate.
References: adequate
Tables: modify as suggested above.
Figures: Please improve image quality for each figure.
Linguistic and typewriting: Linguistic improvement is still nece
Reviewer 2 Report
.
Author Response
Thank you for your comment!